# Variation of Leaf Carbon Isotope in Plants in Different Lithological Habitats in a Karst Area

**Jun Zou [1], Lifei Yu [2,3,\*] and Zongsheng Huang [2]**

1   College of Forest, Guizhou University, Guiyang 550025, China; linzoujun@163.com
2   College of Life Sciences, Guizhou University, Guiyang 550025, China; hzsxjh@126.com
3   Key Laboratory for Mountain Plant Resources Protection of the Education Ministry of China, Guiyang 550025, China
\*   Correspondence: lfyu@gzu.edu.cn or lifeiyugd@163.com; Tel.: +86-13985145271

**Abstract:** Drought is the major factor that limits vegetation recovery in rocky desertification areas. The leaf carbon isotope ($\delta^{13}$C) value is related to plant water-use efficiency (WUE) and is of great significance in revealing the WUE characteristics of species in karst areas. Measurements of the $\delta^{13}$C value in plant leaves and the nutrient and water contents of lithologic soils were obtained for six woody species (cypress, *Cupressus funebris* Endl.; mansur shrub, *Coriaria nepalensis* Wall.; camphor, *Cinnamomum bodinieri* Levl.; birch, *Betula luminifera* H. Winkl.; alder, *Alnus cremastogyne* Burk. and dyetree, *Platycarya longipes* Wu.) planted in three different lithologic soil types (dolomite, dolomite sandstone, limestone) in the karst area of Guizhou Province. The results showed that *C. funebris* in the dolomite sandstone soil had the highest $\delta^{13}$C value (−27.19‰), whereas *C. bodinieri* in the limestone soil had the lowest $\delta^{13}$C value (−31.50‰). In terms of lithology, the average leaf $\delta^{13}$C values were −28.66‰ (dolomitic sandstone), −28.83‰ (dolomite), and −29.46‰ (limestone). The $\delta^{13}$C values of *C. funebris* and *A. cremastogyne* were significantly lower in the limestone soil than in the dolomite and dolomite sandstone soil, indicating that the WUE of some tree species is affected by soil conditions under different lithological development processes. Moreover, the relationship between the $\delta^{13}$C value in the leaves and the comprehensive soil conditions varied among the species, and the $\delta^{13}$C value was negatively correlated with the soil water content in all three soil types. Our study provides basic data on the composition characteristics of the $\delta^{13}$C value of tree species, which is beneficial for the selection of tree species for vegetation restoration and afforestation in karst areas.

**Keywords:** carbon isotope; forest planting; integrated fertility index; lithology; karst area

## 1. Introduction

Karst landforms are characterized by high rates of rock exposure, a thin soil layer, and discontinuous soil cover [1]. Due to the uneven spatial-temporal distribution of precipitation, strong seepage and loss in many karst areas, as well as poor water retention capacity, water is the major factor that limits plant growth in karst areas [2]. Therefore, selecting tree species with high water-use efficiency (WUE) is very important for the success of afforestation in these areas. In 2002, Yu et al. [3] determined the transpiration coefficient of tree species in karst areas to characterize their WUE. Other researchers have measured the WUE of plants using the carbon isotope ($\delta^{13}$C) value. These researchers used spring wheat and desert plants to study the relationship between the $\delta^{13}$C value in the leaves and WUE. The results showed that the leaf $\delta^{13}$C value had a good correlation with the long-term WUE of the plants. The higher the $\delta^{13}$C value, the higher the WUE, which validates the reliability of this index [4–7]. In recent years, much research has been conducted on the relationship between the $\delta^{13}$C value, climate, habitat conditions, and the soil water content. The main trend observed is a higher $\delta^{13}$C value as the

availability of resources decreases [8–12]. However, no research has been reported on the $\delta^{13}$C value and the WUE of plant species from different lithologic habitats in karst areas.

Lithology is a key basic condition that determines a plant's habitat [13]. The composition and proportions of different lithologies are quite different. Limestone mainly consists of $CaCO_3$, whereas dolomite mainly consists of $CaMg(CO_3)_2$. During the lithification process of dolomite, the ratio of the MgO to CaO content increases [14], and the formation and decomposition of Si, Al, $Fe^{2+}$, $Fe^{3+}$, Kr, Ba, and Ni in rock are largely affected by the environment [15,16]. In the course of thousands of years of weathering, rocks of different mineral compositions and structures are affected by multiple factors, such as regional climate and hydrologic factors; therefore, the resulting soil inevitably exhibits differences in the physicochemical properties, soil thickness and nutrient content, among other factors. All of these factors affect plants over a long period of time and thus result in differences in plant C utilization. Studies have shown that different lithologies result in differences in regional water and soil resources [17] as well as hydrological-drought mechanisms [18]. Different water supply statuses will have different effects on plant photosynthetic physiology, resulting in different $\delta^{13}$C values. The distribution of the karst area is very wide; the global karst area accounts for 15% of the total land area. Karst landscapes in southern China cover an area of $54 \times 10^4$ km$^2$, and the area of carbonate rock outcropping in Guizhou is $13 \times 10^4$ km$^2$ [19]. Guizhou is the province with the largest karst area in China, and even the world's hot and subtropical regions [20,21]. Limestone, dolomite and dolomite sandstone account for 42.73%, 17.31% and 6.59%, respectively, of the total carbonate area in Guizhou Province, and they are the largest rock types in the karst area of this province [22].

In this study, six common woody species used for reforestation (cypress, *Cupressus funebris* Endl.; mansur shrub, *Coriaria nepalensis* Wall.; camphor, *Cinnamomum bodinieri* Levl.; birch, *Betula luminifera* H. Winkl.; alder, *Alnus cremastogyne* Burk. and dyetree, *Platycarya longipes* Wu.) planted in three major lithologic soil types (dolomite, dolomite sandstone, limestone) in the karst district of Guizhou Province were selected because they are the dominant species in the plant community. The $\delta^{13}$C values in the leaves of the selected species and the soil water and nutrient contents were measured, which provides a scientific basis for selecting the most suitable tree species for ecological restoration in different lithologic rocky desertification areas.

## 2. Materials and Methods

### 2.1. Study Site

Based on the distribution of carbonate rocks in Guizhou Province, three well-developed sites with three major lithologies (dolomite, dolomite sandstone, limestone) were selected while ensuring that the environmental background was basically the same (Table S1). Of the six species, *C. bodinieri*, *C. funebris*, and *C. nepalensis* occur in all three lithologic soils; *A. cremastogyne* and *B. luminifera* are distributed in dolomite and limestone soils; *P. longipes* grows in the dolomite sandstone and limestone lithologic soils. The details of all sample plots and plant species are shown in Table S1.

### 2.2. Plant Sample Collection

A conventional community survey method was used. Specifically, we chose three samples, each sample area for the tree species was 20 m × 20 m, and each sample area for the shrubs was 10 m × 10 m. In July 2017, fifteen dominant species (Table S1) were collected from three research plots. In the middle and upper part of the crown, outstretched branches were selected and sampled from four directions (east, west, south, north). Twenty fresh mature leaves were collected from each branch. A total of 80 fresh mature leaves were collected and mixed together from each sample woody species. Four sample trees were selected per species, and a total of 60 samples were collected for all 6 species. After the sample collection, the samples were placed in a breathable bag for subsequent drying and grinding prior to the measurements [23].

### 2.3. Soil Sample Collection

Soil samples were collected from the sample plot following an "S" shape, from the 0–20 cm layer, the rhizosphere layer, and the 20–40 cm layer. Every layer of soil samples was collected from five randomly distributed locations in the sample plot and then mixed well. Fifteen dominant species from three sample plots, each with 3–6 replicates, resulted in 90 samples. Subsequently, the soil samples were air-dried under natural conditions, ground, sieved through 0.5-mm and 0.2-mm sieves, and stored in a wide-mouthed bottle.

### 2.4. Measurement Methods

The leaf samples were transported to the laboratory, washed with deionized water, dried at 65 °C in an oven, cooled naturally, crushed by a grinder, and sieved using a 20-mesh sieve before being stored in a sealed bag. Subsequently, the leaf samples were sent to the Stable Isotope Mass Spectrometry Laboratory at the Third Institute of Oceanography, State Oceanic Administration for measurement (Xiamen, China). Isotope analysis was conducted using a thermal conversion elemental analyser-continuous flow isotope ratio mass spectrometry system (TC/EA-IRMS, Delta V Advantage), and the measuring error was <0.05‰.

The $\delta^{13}C$ value was calculated using the international standard form [24]:

$$\delta^{13}C = \left[ \left( \frac{^{13}C}{^{12}C} \right)_{sample} - \left( \frac{^{13}C}{^{12}C} \right)_{standard} \right] \div \left( \frac{^{13}C}{^{12}C} \right)_{standard} \times 1000‰ \tag{1}$$

Here, $(^{13}C/^{12}C)_{sample}$ represents the carbon isotope ratio of the sample, and $(^{13}C/^{12}C)_{standard}$ represents the standard Pee Dee Belemnite (PDB) standard isotope ratio.

The soil physical and chemical properties were analysed using methods in *Soil and Agricultural Chemistry Analysis* [25]. The soil total nitrogen was measured using a distillation method, and the alkali-hydrolysable nitrogen was measured with a diffusion method. The total phosphorus and fast phosphorus contents were determined using the Mo-Sb colorimetric method. The total potassium and Olsen-K contents were determined using flame photometry. The soil organic matter was measured using the potassium dichromate-sulphuric acid-external heating method. The soil water content was determined using the aluminium box drying method, and the soil bulk density was determined using the cutting-ring method.

### 2.5. Integrated Fertility Indices (IFIs)

Nine physicochemical indices that are closely associated with soil characteristics were used to characterize the lithological soil conditions. The comprehensive condition indices of soil were analysed using principal component analysis (PCA) [26,27]. The variance contribution rate of each principal component $Y_j$ is regarded as the weight. The calculation formula is as follows:

$$IFIs = \sum_{j=1}^{2} a_j y_j \tag{2}$$

Here, $a_j$ represents the variance contribution rate of the *j-th* principle component, and $y_j$ represents the score of the *j-th* principle component.

### 2.6. Data Processing

The data were analysed using Excel 2013 and SPSS 19.0 software (IBM Corp., Chicago, United States). The leaf $\delta^{13}C$ values among the lithology types and different tree species were analysed using one-way analysis of variance (ANOVA). The relationship between the leaf $\delta^{13}C$ values and comprehensive soil conditions and soil water content was analysed by correlation analysis (CA,

significance level: $\alpha = 0.05$). The soil physical and chemical values are based on the average values in different soil profile levels in the sample plots.

## 3. Results

### 3.1. $\Delta^{13}C$ Values of the Dominant Species in the Three Lithological Soil Types

The $\delta^{13}C$ values of dominant species in the three lithological soils were determined (Table 1). The results showed that *C. funebris* planted in the dolomite sandstone soil had the maximum $\delta^{13}C$ value (−27.19‰), whereas *C. bodinieri* had the minimum $\delta^{13}C$ value (−31.50‰) in the limestone soil. The plant $\delta^{13}C$ values across the three lithological soil types were −28.66‰ (dolomite sandstone), −28.83‰ (dolomite), and −29.46‰ (limestone). Moreover, the $\delta^{13}C$ value of *C. funebris* in the limestone soil was significantly different from that in the other lithological soil types ($p < 0.001$). There was a significant difference in the $\delta^{13}C$ value of *A. cremastogyne* in limestone and dolomite soil ($p = 0.024$). Except for *C. funebris* and *A. cremastogyne*, the four other species did not exhibit significant differences in the $\delta^{13}C$ values among the different lithological soil types.

**Table 1.** Characteristics of the $\delta^{13}C$ values of the dominant species in three lithological soil types.

| Lithology | Dominant Species | Minimum Value (‰) | Maximum Value (‰) | Mean (‰) ± Standard Deviation | Significant Difference Across sites |
|---|---|---|---|---|---|
| Dolomite | *Cinnamomum bodinieri* | −30.70 | −29.97 | −30.37 ± 0.36 [a] | − |
| | *Cupressus funebris* | −27.60 | −27.26 | −27.45 ± 0.15 [d] | 1 |
| | *Coriaria nepalensis* | −28.10 | −27.70 | −27.88 ± 0.17 [d] | − |
| | *Betula luminifera* | −29.94 | −28.96 | −29.55 ± 0.47 [b] | − |
| | *Alnus cremastogyne* | −29.40 | −28.39 | −28.91 ± 0.41 [c] | 1 |
| Dolomite sandstone | *Cinnamomum bodinieri* | −32.49 | −30.10 | −31.08 ± 1.23 [a] | − |
| | *Cupressus funebris* | −27.88 | −26.20 | −27.19 ± 0.76 [c] | 1 |
| | *Coriaria nepalensis* | −28.14 | −26.90 | −27.51 ± 0.60 [c] | − |
| | *Platycarya longipes* | −29.62 | −28.30 | −28.87 ± 0.57 [b] | − |
| Limestone | *Cinnamomum bodinieri* | −32.00 | −30.44 | −31.50 ± 0.72 [a] | − |
| | *Cupressus funebris* | −29.80 | −29.32 | −29.51 ± 0.21 [b] | 2 |
| | *Coriaria nepalensis* | −28.60 | −26.99 | −27.65 ± 0.77 [c] | − |
| | *Betula luminifera* | −29.80 | −29.25 | −29.58 ± 0.27 [b] | − |
| | *Alnus cremastogyne* | −30.00 | −29.26 | −29.69 ± 0.31 [b] | 2 |
| | *Platycarya longipes* | −29.90 | −28.20 | −29.38 ± 0.80 [b] | − |

The letters represent significant differences in the $\delta^{13}C$ values of different species within each site. The numbers represent significant differences in the $\delta^{13}C$ values across sites for each specie.

### 3.2. $\Delta^{13}C$ Values of Different Dominant Species

A comparison of the $\delta^{13}C$ values of different dominant plants indicated that *P. longipes*, *A. cremastogyne* and *B. luminifera* (−29.13‰, −29.30‰ and −29.57‰) show significantly lower mean $\delta^{13}C$ value than *C. nepalensis* and *C. funebris* (−27.68‰ and −28.05‰, $p < 0.001$), and significantly higher mean $\delta^{13}C$ value than *C. bodinieri* (−30.98‰, $p < 0.001$) across the three lithologies (Table 1). In the dolomite soil, there was an extremely significant difference in the $\delta^{13}C$ value between *C. bodinieri*, *C. funebris,* and *C. nepalensis* as well as between these three species and the other species ($p \leq 0.001$), whereas the difference in the $\delta^{13}C$ value between *C. funebris* and *C. nepalensis* was not significant ($p = 0.115$). In the dolomite sandstone soil, there was an extremely significant difference in the $\delta^{13}C$ value between *C. bodinieri* and the other species ($p < 0.001$); there was no significant difference in the $\delta^{13}C$ value between *C. funebris* and *C. nepalensis* ($p = 0.589$), whereas there was a significant difference in the $\delta^{13}C$ value between these two species and the other species ($p = 0.040$ or $p = 0.014$). In the limestone soil, there was an extremely significant difference in the $\delta^{13}C$ value between *C. bodinieri* and *C. nepalensis* and the other species ($p < 0.001$), but there was no significant difference in the $\delta^{13}C$ value between *B. luminifera*, *A. cremastogyne*, and *P. longipes* ($p = 0.780$, $p = 0.450$ and $p = 0.631$). Furthermore, the mean $\delta^{13}C$ values of *C. nepalensis*, *C. funebris* and *C. bodinieri* (the three species occurring in all three lithological soil types) in limestone soil (−29.55‰) were lower than in dolomite and dolomite sandstone soil types (−28.57‰ and −28.59‰, $p = 0.168$ and $p = 0.180$).

### 3.3. Relationship between the $\Delta^{13}C$ Value of Dominant Species and Comprehensive Lithological Soil Conditions

To investigate the relationship between the $\delta^{13}C$ value in the plant leaves and the conditions of the different lithological soils, nine physicochemical indices that are closely associated with soil characteristics were used to characterize the lithological soil conditions (Table 2). The average value of each index in the table is standardized and processed. PCA was performed to evaluate the comprehensive soil conditions. The results indicated that the principal components $y_1$ and $y_2$ accounted for 74.092% and 25.908%, respectively, of the total variation. The linear equation and scoring matrix of the two principal components are as follows, and the details are shown in Table 2:

$$y_1 = -0.497x_1 + 0.663x_2 + 0.977x_3 + 0.960x_4 + \cdots + 0.897x_9 \tag{3}$$

$$y_2 = 0.868x_1 - 0.748x_2 + 0.215x_3 + 0.281x_4 + \cdots + 0.442x_9 \tag{4}$$

**Table 2.** Characteristics of the three lithological soil types.

| | | Soil Bulk Density ($x_1$, g/cm³) | Soil Water Content ($x_2$, %) | Organic Matter ($x_3$, g/kg) | Total Nitrogen ($x_4$, g/kg) | Total Phosphorus ($x_5$, g/kg) | Total Potassium ($x_6$, g/kg) | Alkali-Hydrolysable Nitrogen ($x_7$, g/kg) | Available Phosphorus ($x_8$, mg/kg) | Olsen-K ($x_9$, mg/kg) |
|---|---|---|---|---|---|---|---|---|---|---|
| Dolomite | $\bar{x}$ | 0.998 | 29.610 | 55.671 | 2.889 | 0.467 | 17.797 | 183.255 | 2.087 | 81.923 |
| | $x'$ | −1.068 | 0.956 | −0.100 | −0.178 | −0.510 | 0.809 | −0.023 | 0.237 | −0.372 |
| Dolomite sandstone | $\bar{x}$ | 1.078 | 23.280 | 47.991 | 2.453 | 0.424 | 7.120 | 143.903 | 1.639 | 76.700 |
| | $x'$ | 0.914 | −1.039 | −0.946 | −0.899 | −0.642 | −1.118 | −0.988 | −1.097 | −0.761 |
| Limestone | $\bar{x}$ | 1.047 | 26.840 | 66.088 | 3.650 | 1.018 | 15.028 | 225.450 | 2.296 | 102.105 |
| | $x'$ | 0.154 | 0.083 | 1.046 | 1.077 | 1.152 | 0.309 | 1.011 | 0.860 | 1.133 |

$\bar{x}$: mean value; $x'$: The standardized value; $x'_{ij} = \frac{x_{ij} - \overline{x_j}}{S_j}$ $S_j = \sqrt{\frac{1}{n-1} \sum_{i=1}^{n} \left( x_{ij} - \overline{x_j} \right)^2}$.

According to the formula of *IFIs*, the ranking based on the fertility of the three lithological soil types was limestone (4.954), dolomite (0.047), and dolomite sandstone (−5.001), indicating that limestone contained the most nutrients, followed by dolomite and dolomite sandstone (Table 3). The relationship between the $\delta^{13}C$ values of *C. funebris*, *C. bodinieri*, and *C. nepalensis* and the comprehensive value (*IFIs*) of the soil condition was then analysed (Figure 1). The results suggested that the $\delta^{13}C$ values of the three species were negatively correlated with the comprehensive soil condition. The $\delta^{13}C$ value of *C. funebris* had a significant negative correlation with the comprehensive soil condition ($R^2 = 0.852$, $p < 0.001$), inferring that a lower $\delta^{13}C$ value is related to a better comprehensive soil condition. The $\delta^{13}C$ values of *C. bodinieri* and *C. nepalensis* were negatively correlated with the comprehensive soil condition and the soil water content to various degrees. However, the correlation coefficient was 0.426 ($p = 0.0018$) and 0.624 ($p = 0.0022$) for *C. bodinieri* and *C. nepalensis*, respectively, implying that the negative correlation between the $\delta^{13}C$ values of these two species and the comprehensive soil condition was lower than *C. funebris*.

**Table 3.** Scores and ranks of the principal components of the soil indices.

| | **P₁** | | **P₂** | | **Overall Evaluation** | |
|---|---|---|---|---|---|---|
| **Lithology Type** | $y_1$ | **Rank** | $y_2$ | **Rank** | *IFIs* | **Rank** |
| Dolomite | 0.997 | 2 | −2.669 | 3 | 0.047 | 2 |
| Dolomite sandstone | −7.111 | 3 | 1.033 | 2 | −5.001 | 3 |
| Limestone | 6.114 | 1 | 1.636 | 1 | 4.954 | 1 |

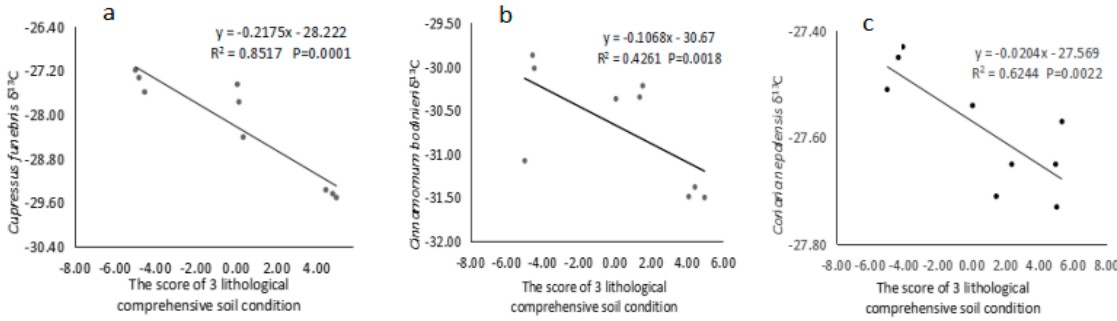

**Figure 1.** Relationship between the $\delta^{13}$C value in the plant leaves and the comprehensive lithological soil condition. (**a**) *Cupressus funebris*, (**b**) *Cinnamomum bodinieri*, (**c**) *Coriaria nepalensis*.

### 3.4. Relationship between the $\Delta^{13}$C Value of the Dominant Species and the Soil Water Content

The $\delta^{13}$C value in leaves is closely related to the plant WUE. Therefore, we speculated that there was a correlation between the $\delta^{13}$C value and the soil water content. By analysing the relationship between these variables (Figure 2), we found that the $\delta^{13}$C values of the dominant species were negatively correlated with the soil water content, i.e., the higher the soil water content, the lower the $\delta^{13}$C value. For the dolomite soil type ($R^2 = 0.8293$, $p = 0.0002$), the negative correlation between the soil water content and the leaf $\delta^{13}$C values was higher than limestone ($R^2 = 0.403$, $p = 0.0013$) and dolomite sandstone ($R^2 = 0.393$, $p = 0.0012$) soil types.

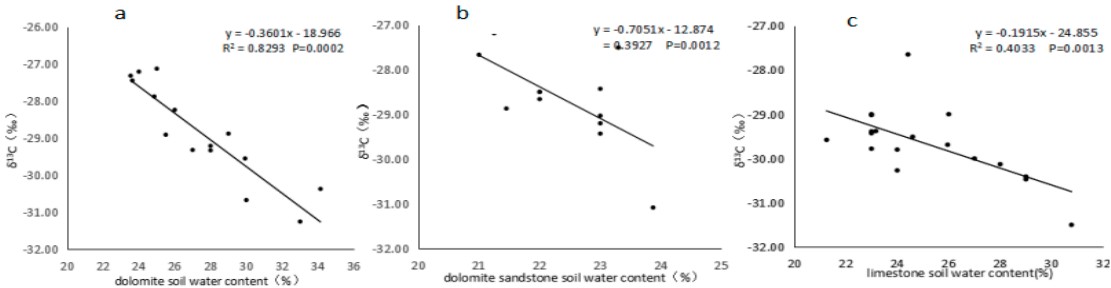

**Figure 2.** Relationship between the $\delta^{13}$C value of the dominant species and the soil water content in different lithological soil types. (**a**) dolomite, (**b**) dolomite sandstone, (**c**) limestone.

## 4. Discussion

According to the 2013–2017 meteorological data released by the Guizhou Provincial Meteorological Administration (Table S2), the dolomite sandstone soil showed the highest mean temperature, followed by dolomite and limestone soils. Consistent with the changes in temperature, our results showed that *C. funebris* in the dolomite sandstone soil had the highest $\delta^{13}$C value (−27.19‰), whereas *C. bodinieri* had the smallest $\delta^{13}$C value (−31.50‰) in the limestone soil. This is similar to the $\delta^{13}$C range of plants in the Guizhou karst mountain area (−26.98‰~−29.15‰) [19] and that of woody plant species in karst desertification areas (−25.55‰~−30.28‰) [28], far above the maximum values (−27.60‰~−38.65‰) of rainforests plants in Xishuangbanna, Yunnan [29], and far below the minimum values (−23‰~−29‰) of desert plants [30] and deciduous broad-leaved trees (−22.23‰~−31.38‰) [31,32]. These studies suggest that variations (from small to large) in the $\delta^{13}$C values of plants in different climatic zones are correlated with hydro-thermal gradient changes (from south to north). Carbon isotopic ratios on plant tissues depend on atmospheric $CO_2$ composition, stomatal conductance and photosynthetic ratio among other parameters [33]. Thus, temperature and water are key factors that influence the $\delta^{13}$C value of species. In this study, the $\delta^{13}$C value of *C. funebris* in the limestone soil was significantly different from that in other lithological soil types ($p < 0.001$), and the $\delta^{13}$C value of *A. cremastogyne* differed significantly between the limestone soil and dolomite soil ($p = 0.024$). We speculated the reason for this result is that dolomite and dolomite sandstone soils have more similar climate characteristics

than limestone soil (temperature: Table S2; precipitation: Table S1). Moreover, carbon isotopic ratio mainly depends on WUE, and thus different soils resource conditions will affect isotopic composition in term of water availability to plants [34]. Dolomite and dolomite sandstone soils have different structures but share a similar chemical composition. Therefore, the two lithologically developed soils have the same effect on the $\delta^{13}C$ values of plants. This may have resulted in the lack of a significant difference in the $\delta^{13}C$ value of *C. funebris* between the dolomite and dolomitic sandstone.

The $\delta^{13}C$ value of plants is a result of plant and environmental conditions [32,35]. Plants of the same species or functional group may have different $\delta^{13}C$ values under different environmental conditions. Our research showed that the $\delta^{13}C$ value had a good correlation with the soil comprehensive index (*IFI*) for six plant species, but some of the correlations were not strong, indicating that the $\delta^{13}C$ values of different plant species in the karst area were inconsistent with respect to different comprehensive soil conditions. Previous studies have shown that plants normally have lower $\delta^{13}C$ values when sufficient resources are available in the environment [5,36]. Furthermore, other studies have noted a negative correlation between $\delta^{13}C$ values in leaves and the soil water content for some tree species, and plants growing in drought habitats have higher leaf $\delta^{13}C$ values [32,37]. Here, we found the mean $\delta^{13}C$ values of *C. funebris*, *C. bodinieri*, and *C. nepalensis* in limestone soil were lowest, followed by dolomite and dolomite sandstone soil types. This result was consisted with the comprehensive limestone soil condition better than the other soil lithologies. However, for some lithological soil types, expect for the dolomite soil, the soil water content was not significantly correlated with the leaf $\delta^{13}C$ values. It is possible that the main fractionation processes occurring at leaf level play an important role in the carbon isotopic ratios. When a tree is subjected to water stress, the stomatal conductance and photosynthetic rates of the plant leaves are affected, and there are different fractionation effects on the isotopes [33]. Therefore, the leaf $\delta^{13}C$ value had a strong negative correlation with the soil water content. The mean annual precipitation at the dolomite sandstone (1240.4 mm) and limestone (1430.9 mm) soils was higher than the dolomite soil (1100 mm, Table S1), resulting in their leaf $\delta^{13}C$ values not being significantly correlated with the soil water content.

Common tree species growing at different soil depths (or under different nutrient conditions) in rocky desertification areas normally exhibit two types of responses with regard to the $\delta^{13}C$ value. First, the $\delta^{13}C$ value has been shown to change non-significantly for species such as *Acer buergerianum* and *Mallotus repandus*; second, the $\delta^{13}C$ value decreases with increasing soil depth for species such as *Mallotus philippinensis* and *Pistacia weinmannifolia*; the latter response is the dominant mode [38]. In deep soils with poor nutrient conditions, the $\delta^{13}C$ values of plants in karst mountainous areas have shown both positive and negative responses [19]. Research on the correlation between the $\delta^{13}C$ value of *Quercus fabri* and soil nitrogen, phosphorus, and potassium levels in karst areas [23] has shown that the $\delta^{13}C$ value was positively correlated with the levels of alkali-hydrolysable nitrogen and available potassium, negatively correlated with the Olsen *p* value, and highly negatively correlated with the comprehensive soil condition ($R^2 = 0.9779$). We suspected that the possible reason for these research differences is the association between $\delta^{13}C$ value and soil properties is indirect, and climate characteristics need to be considered. Our previous results suggested that the $\delta^{13}C$ values of three plant species show varying degrees of negative correlation with the comprehensive soil conditions and soil water contents, but they are also related to temperature and precipitation. Moreover, other researches demonstrated a tighter link between $\delta^{13}C$ value and soil depth than with climate conditions in *Pinus sylvestris*, because of the amount of water available to roots [38,39]. This inferred that under well-watered and mild temperature conditions, where soil water retention capacity is low, soil properties related to an enhance water retention should be the key factor conditioning carbon isotopic composition on plant tissues.

In summary, the $\delta^{13}C$ method could be used to determine the comprehensive soil conditions in karst areas. This method is advanced, simple and worthy of exploration. However, the association between $\delta^{13}C$ value and soil properties is indirect, and some problems remain to be addressed. A large number of experimental studies are required on the sampling method, control conditions, specific

indicators, and the effects of the climate. In addition, some tree species exhibit more sensitivity regarding the $\delta^{13}$C value, and some have a large ecological range; therefore, the selection of tree species requires careful consideration. Determining whether the $\delta^{13}$C value corresponds to the changes in soil conditions or water stress is the greatest obstacle in characterizing soil and water conditions using $\delta^{13}$C values. Therefore, more theoretical and technical breakthroughs are needed to reveal the comprehensive soil conditions in karst areas using the $\delta^{13}$C value in plant leaves.

**Supplementary Materials:** The following are available online at http://www.mdpi.com/1999-4907/10/4/356/s1, Table S1: Sample plot characteristics of forest stands across three lithological soil types. Table S2: The mean temperature in five years of sample plots across three lithological soil types.

**Author Contributions:** L.Y. conceived and designed the project. J.Z. conducted the experiments and wrote the manuscript. J.Z. and Z.H. analysed the data. All authors read and approved the final manuscript.

**Acknowledgments:** This work was supported by a project of the National Key Research and Development Program of China (2016YFC0502604), the Guizhou Province major special project ([2014]2002), and the National Natural Science Foundation (31560187).

**Conflicts of Interest:** The authors declare no conflicts of interest.

## Abbreviations

The following abbreviations are used in this manuscript:

| | |
|---|---|
| WUE | Water-use efficiency |
| IRMS | Isotope ratio mass spectrometry |
| PDB | Pee Dee Belemnite standard |
| PCA | Principal component analysis |
| CA | Correlation analysis |
| IFI | Integrated fertility index |

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
