# Peer review of "Variation of Leaf Carbon Isotope in Plants in Different Lithological Habitats in a Karst Area"

_forests, doi:10.3390/f10040356_

Round 1
Reviewer 1 Report
Dear Authors
The manuscript main aim is to test whether δ13C leaf values on tree and shrub species growing on lithological different karstic soils is a valid parameter to improve the selection of reforestation species in karstic areas rehabilitations, and if δ13C is a good proxy of soil properties. To do so, a nice record of δ13C in leaves of 6 species with different leave morphology and phenology was taken in karstic regions showing interesting values.
Although the main hypothesis behind this article (δ13C leaf values relate to water use efficiency and that should be linked to soil properties in karstic areas) is original and sound, it is not well developed in the manuscript, and the design of the experiment does not account for the main processes driving carbon fractionation, that is to say, stomatal conductance and photosynthetic rate. Perhaps climatic stability added to different soil water retention capacities could imply that soil water availability drives leave fractionation processes. However this is not discussed or analized and therefore it is not possible to conclude in which amount the soil properties where affecting the δ13C leaf signature. Both stomatal conductance and photosynthetic rate are processes where a sum of different environmental and physiological features are involved, being soil water content or soil fertility two of them. On the other hand, there has been a lot of effort in the sampling and it is a good database where to analyze the how leaf δ13C varies in the 6 studied an how can soil properties might be involved in that. A better statistical analysis performance (maybe using mixed effect models), adding climatic data could improve the understanding about how soil properties could affect the isotopic signature of leaves. The discussion is quite speculative as main reasoning relays on other studies but not in their own results, trying to scale up their findings to global scale in karstic areas. A better approach would be to focus on the obtained results, including climatic data ( e.g gridded data as CRU dataset (https://crudata.uea.ac.uk/cru/data/hrg/) or local meteorological stations). With this data it might be better explained water use efficiency seasonality in studied locations, and linked to soil properties.
Regarding the text itself, the introduction does place the study into a context, and it briefly review the state of the research in this field. Sampling design is not straightforward. Results are clear, but discussion is not well referenced (point studies even in other continents with different species from the present study are used instead of broader single studies or several studies showing same patterns in different species or locations). Bibliography is not following the journal guides.
Line 63-65: This two sentences are lacking a reference.
Line 75-76: The water use efficiency is not determided in this study as stated here.
Line 81: As part of the material and methods, some words about the different lithologies should be mentioned, or at least the names of the different lithologies. In my opinion it would be useful also to have some words where the three lithologies are sorted by their water retention capacity, or some other hydraulic parameter that enables the reader to better understand where to expect the strongest drought stresses
Line 83-85: As part of material and methods, I would expect to see here the full name of all the six analyzed species, not only the 3 most abundant ones. It would be also usefull a short description of their leaf characteristics as later on in the discussion authors reffer to them and the reader has not learned before which leave morphology has each species.
Line 93-94: I might have misunderstood but, for me it is not easy to know how can authers get 60 samples, if four sample trees were selected per specie and there were 6 species in 3 plots (4*6*3= 72). I would suggest to review this numbers, or explain it in a way that is easier to understand.
Line 98: Review the sentence, there is an "in" that should be removed
Lines 99-101: I lost again the track of the number of samples. First it is stated that 5 random distributed samples from each of the two layers were mixed together. But then it is stated that six samples were collected for each tree (meaning the 4 trees sampled before?) for the 6 species; but then 6*4*6= 144, plus the samples that are mentioned in the previous lines. I would suggest to better explain the soil sampling collection.
Lines 86-103 : It would be useful to know when the samples were taken and if possible also at which stage (was the leave still forming and elongating or was it already mature at its final stage).
Line 147 (Table 1 column titles): formatting makes it a bit difficult to read the titles, as in number of samples, of is just next to the following column
Line 147 (Table1 footnote): First sentence of the legend does not seem to be correct, letters don't represent significant differences but just different species.
Line 147 and 162 (Table 1 and 2): I would suggest to sort the species in the same order in both tables to make easier for the reader to compare (e.g. in Table 2 use the same species order as in Table 1 on the Limestone lithology rows).
Line 186-188: The indicated P-values do indicate a significant correlation. R2 values might be lower than in C. funebrensis, but still the negative correlation is quite clear. In any case it might be also useful to show the correlation coefficient values.
Line 200-201: This sentence does not make too much sense. In the previous lines authors showed hoe δ13C of the dominant species correlated with water soil content values in the 3 different lithologies.
Lines 207-229: This section of the discussion is speculative. Carbon isotopic ratios on plant tissues depend in first instance on atmospheric CO2 composition, stomatal conductance and photosynthetic ratio among other parameters (see Gessler et al 2014). Here authors discuss how carbon isotopic composition can vary depending on lithological soil carachteristics. This might be truth, but if linked to water contents of the different lithologies and also to their different climates. Furthermore, authors compare their results with species and location far away from their experiment ranges beyond the limits of their analyzed results.
Line 211-212: values are within the values of the compared rain forests and desert plants. I would suggest to rephrase in a similar way to "far above the maximum values of rain forests plants " instead of "higher" and "far below the minimum values of desert plants" instead of "lower".
Line 225-226: Reference would be needed for this statement. Carbon isotopic ratio mainly depends on water use efficiency, thus different soils resource conditions will affect isotopic composition in term of water availability to plants.
Line 231-244: This section of the discussion contain mistakes on the results interpretation. First sentence of the discussion is not supported by the results. The classification authors make is not in agreement with table 2 values. Thus the further discussion is not correct. At the end of the paragraphs the authors suggest some species to be used in vegetation restoration based not only in water use efficiency but in other parameters (higher biomass production and improvement of habitats) that are not studied in the present work.
Lines 251-254: It is discussed that A. cremastogyne has different values in dolomite sandstone than in limestone, but this specie was not sampled at dolomite sandstone locations. It is also stated that limestone soil condition is better than the other soil lithologies based in the results in one out of six species.
Line 260-263: The reasoning here is not clear. First it is stated that there is higher rainfall (not quantified) thus there is not water stress. But afterwards the authors state that δ13C differences are depend on water stress.
Lines 265-285: In this section of the discussion the authors discuss the use of δ13C as a proxy of soil properties. However authors miss again the inclusion of climatic variables in the discussion. Under well watered and temperature mild conditions, where soil water retention capacity is low, soil properties related to an enhance water retention should be the key factor conditioning carbon isotopic composition on plant tissues. The main fractionation processes occurring at leaf level (due to stomatal conductance and photosynthetic rates) are missing in this section and generally in the manuscript, just focusing on soil properties without reasoning why soil properties might be driving plant carbon isotopic composition.
Bibliography
Gessler A, Pedro Ferrio J, Hommel R, Treydte K, Werner RA, Monson RK (2014) Stable isotopes in tree rings: towards a mechanistic understanding of isotope fractionation and mixing processes from the leaves to the wood. Tree Physiology. 34:796-818.
Author Response
Thanks for your comments, which have greatly benefited our manuscript. We have made thorough revisions to the article, especially in the Discussion and Reference sections. Moreover, we provided a point-by-point response to your comments!

Reviewer 2 Report
Dear authors,
I revised the new version of your manuscript on ‘Variation of leaf d13C in plants in different lithological habitats in a karst area’. The manuscript has improved significantly. The methodology, results and main conclusions are now clearer. The text is more concise and better organized.
Some comments follow below:
L32: replace ‘tree species’ by, e.g. ‘woody plant species’, ‘woody species’ or ‘plant species’ to integrate both trees and shrubs along the manuscript.
L42: ‘to characterize/describe their WUE´?
L54: delete ‘from low to high’
L65: what do you mean with ‘the most developed province in China, and even the world’? Please, clarify it.
L64-69: There is no need to give these details about the area of study, at least in the Introduction. I would rather place the study in a global context, which would additionally increase the impact of your study.
L82: ‘The overall stand characteristics were dominated by the characteristics of the selected dominant tree species.’ It is not clear what you mean. Please, clarify it or delete this sentence.
L240-243: Please, soften this sentence. Your study supports these different aspects rather than indicates.
L253: ‘better’ in terms of what? Please, specify in the text. Same with ‘worsening’ in L258.
L262: Add a reference or explain the ‘fractionation effects on the isotopes’.
L259: This paragraph is confusing. For example, you mention that ‘the soil water content was not significantly correlated with the leaf δ13C values’ (L259) and ‘the leaf δ13C value had a strong correlation with the soil water content’ (L262). Please, clarify it.
Table 1. It seems letters do not indicate significant differences but different species, while the numbers associated to the different letters indicate significant differences. Please, clarify it in the table footer.
Table 2: Both in Table 1 and 2, it is not necessary to mention that ‘The same letter represents no significant difference in the δ13C value, whereas different letters indicate a significant difference’, since it is inferred from the previous sentence (‘a, b, and c within a column represent significant differences in the δ13C values of different species’).
Author Response

(The authors gave the same response as above.)
